# Effect of Facial Skin Temperature on the Perception of Anxiety: A Pilot Study

**DOI:** 10.3390/healthcare8030206

**Published:** 2020-07-09

**Authors:** Elba Mauriz, Sandra Caloca-Amber, Ana M. Vázquez-Casares

**Affiliations:** Department of Nursing and Physiotherapy, Universidad de León, Campus de Vegazana, s/n, 24071 León, Spain; scaloa00@estudiantes.unileon.es (S.C.-A.); ana.vazquez@unileon.es (A.M.V.-C.)

**Keywords:** infrared thermal imaging, facial temperature, stress, anxiety, simulation-based learning

## Abstract

The extent of anxiety and psychological stress can impact upon the optimal performance of simulation-based practices. The current study investigates the association between differences in skin temperature and perceived anxiety by under- (*n* = 21) and post-graduate (*n* = 19) nursing students undertaking a cardiopulmonary resuscitation (CPR) training. Thermal facial gradients from selected facial regions were correlated with the scores assessed by the State-Trait Anxiety Inventory (STAI) and the chest compression quality parameters measured using mannequin-integrated accelerometer sensors. A specific temperature profile was obtained depending on thermal facial variations before and after the simulation event. Statistically significant correlations were found between STAI scale scores and the temperature facial recordings in the forehead (r = 0.579; *p* < 0.000), periorbital (r = 0.394; *p* < 0.006), maxillary (r = 0.328; *p* < 0.019) and neck areas (r = 0.284; *p* < 0.038). Significant associations were also observed by correlating CPR performance parameters with the facial temperature values in the forehead (r = 0.447; *p* < 0.002), periorbital (r = 0.446; *p* < 0.002) and maxillary areas (r = 0.422; *p* < 0.003). These preliminary findings suggest that higher anxiety levels result in poorer clinical performance and can be correlated to temperature variations in certain facial regions.

## 1. Introduction

Clinical simulation training enables students to put into practice classroom knowledge. By reducing the gap between theoretical knowledge and real practice, healthcare practitioners prepare to manage real demands during direct patient care while minimizing the risks derived from an inexperienced practice [1,2]. Simulation-based learning carried out in highly realistic scenarios also promotes the development of technical and non-technical skills, such as critical thinking, self-confidence and emotion control [3,4,5,6].

In spite of these inherent advantages, the number of tools for the objective quantification of the competences acquired throughout simulation training is still scarce. Most of the research in this field is aimed at developing medical simulators capable of integrating digital measurements [7]. However, the performance assessment of simulation-based practices still remains a challenge since clinical outcomes may be affected by the feelings and emotions of participants [8,9]. In particular, increased stress and anxiety levels may impair the simulation performance by negatively affecting attention and decision making [10,11,12,13,14]. Thus, anxiety can be associated with attention deficits and memory impairments, thereby diminishing the cognitive capacity during executive functions, especially among young adults under antidepressant therapy [15]. Psychological stress and anxiety may also interfere with critical thinking and self-efficacy, resulting in poor clinical performance, especially in vital emergency situations [16,17]. Therefore, measuring psychological stress and anxiety throughout valid and reliable instruments is essential for assessing the simulation performance [3,5,18].

To overcome this problem, a variety of methods including stress and anxiety scales, pre- and post-simulation self-reports, vital signs monitoring and analysis of cortisol levels have been already employed [3,5,17]. However, the extent of anxiety and physiological stress has been rarely measured through straightforward robust methods. Only a few studies, involving non-conventional techniques, have been used for identifying the influence of these domains on the learning outcomes [3,18,19,20]. Amongst them, eye tracking and thermal imaging technologies have been recently applied to assess optimal performance during simulation [11].

In particular, infrared thermography (IRT) has proved its usefulness as a non-invasive technique for monitoring biomedical events such as face thermoregulation [21,22]. IRT can take advantage of the infrared fraction of electromagnetic radiation emitted by the human skin for detecting human emotion and cognitive load perception. The physiological activation of a specific facial area yields an increase in temperature due to the rise in blood perfusion, whereas diminished temperatures corresponding to low physiological activation indicate less facial irrigation [23]. The variation in skin temperature can be measured by thermographic cameras capable of providing thermograms of facial heat distribution that can be associated with emotions and physio-psychological states. When monitoring stress and anxiety through thermal facial variations, the most critical areas to take into consideration are the nose, mouth, cheeks, forehead, periorbital and maxillary regions [19,24]. Therefore, it is of interest to examine whether the temperature pattern obtained by combining temperature changes from the facial regions of interest may be correlated with feelings and emotions in stressful situations, such as simulation training environments [19,25].

The aim of this pilot study was to examine the relationship between differences in skin temperature of distinct facial regions and the stress and anxiety perceived by bachelor and master nursing students during a cardiac arrest simulated based-scenario. Our hypotheses were that (i) thermal facial gradients obtained by infrared thermal imaging can be associated with the scores assessed by stress and anxiety validated questionnaires and (ii) higher anxiety levels will result in poorer clinical performance with regard to temperature variations in certain facial regions.

## 2. Materials and Methods

### 2.1. Study Design

This study used a pre-test–post-test quasi-experimental design. A convenience sample was selected among second-year bachelor of nursing students (BS, *n* = 21) and post-graduate registered nurses undertaking studies in a master of science in advanced clinical nursing (MS, *n* = 19) from a mid-sized Spanish university.

There were no other exclusion criteria than being included in their respective academic programs, that is, second-year bachelor’s and master’s degrees. Both under- and post-graduate students had received CPR theoretical classes and were familiar with low-fidelity simulation mannequins. The recruitment process was developed during the lecture time and via online postings. The participation was voluntary and did not include remuneration or course credit.

Ethical approval (ETICA-ULE-004-2019) was obtained from the University’s Institutional Review Board (Ethics Committee, University of León). All participants signed the informed written consent form in accordance with the Helsinki Declaration guidelines. The anonymity and confidentiality of the data were guaranteed.

### 2.2. Data Collection and Instruments

Participants were exposed to a cardiac arrest simulated environment comprising the performance of cardiopulmonary resuscitation (CPR) maneuvers according to ERC (European Research Council) guidelines for 2 consecutive minutes. Data regarding CPR quality parameters (compression release, depth and rate, ventilation volume, number of compressions/ventilations/cycles) were collected from a medium fidelity mannequin (Little Anne QCPR™, Laerdal Medical, Stavanger, Norway) and extracted using the software provided by the manufacturer (QCPR Learner Mobile Application™, Laerdal Medical, Stavanger, Norway). This information was used to determine the performance scores. The students were instructed to participate individually in the simulation scenario and did not receive guidance during the study.

Prior to the intervention, all the participants completed a self-administered sociodemographic and knowledge questionnaire about basic life support (BLS). Physiological values including heart rate, blood pressure and oxygen saturation were measured immediately prior and following the simulation practice (data not shown), whereas stress and anxiety traits were also self-rated using validated scales at pre- and post-simulation moments. A 100 mm visual analogue scale (VAS) was used to evaluate stress levels by measuring the amount of stress experienced from 1 (very little) to 10 (very much). The anxiety traits were self-assessed by the State-Trait Anxiety Inventory (STAI) questionnaire comprising 20 statements about the temporary state or emotional condition perceived by the participants, with 4 gradual response options regarding the level of agreement or disagreement with the stated provision [5,26].

Immediately prior and following CPR performance, thermographic photographs corresponding to the selected facial regions of interest were taken using an infrared camera (FLIR E6, FLIR Systems, Inc., Wilsonville, OR, USA). The thermographic camera had an image resolution of 160 × 120 (19,200 pixels), a measurement range from −20 °C to 250 °C and an accuracy for ambient temperature of ±2%.

The set of images were obtained by the same observer to avoid interexaminer variation in a closed room of 40 m^2^ equipped with neon lights and low incidence of natural light. Temperature and relative humidity were constantly monitored throughout the whole duration of the simulation event. The camera was positioned in front of the face of participants and the distance between the camera and the participants was fixed at 1 m. Thermographic images were classified according to the moment in which they were taken, prior (pre-test) or after (post-test) the simulation procedure.

The same facial region comprising head, neck and the upper part of the thorax, approximately above the clavicular area were considered for each participant. Specifically, one point and four regions of interest with identical rectangular size were selected: nose point and forehead, periorbital, maxillary and neck regions, as previously described [27,28].

To perform the analysis of the thermographic images, the minimum, average and maximum temperature values were considered for each selected region. The temperature difference between pre-test and post-test thermal values was also calculated subtracting the subsequent temperature minus the previous one. All the thermographic images were processed using the software provided by the camera’s manufacturer (FLIR TOOLs Software Version 1.1, FLIR Systems, Inc., Wilsonville, OR, USA).

### 2.3. Data Analysis

Descriptive statistics were used for sociodemographic characteristics, continuous variables were expressed as means (Standard Deviation: SD) and categorical variables as absolute numbers and percentage. Normality in the distribution of data was assessed using the Kolmogorov–Smirnov test. Independent t tests, chi-square and Mann–Whitney U tests were used as appropriate to compare the differences between bachelor and master nursing students. Comparison between pre-test and post-test simulation scores was completed using Student’s t test paired and Wilcoxon-signed rank test for normal and non-normal distribution variables, respectively. Bivariate correlations between thermographic values and the performance scores variables were analyzed using the Pearson or Spearman statistics. Several multiple linear regression models were tested, considering STAI and CPR scores as dependent variables and temperature recordings of the selected facial regions as independent variables. The software package SPSS for Windows version 25.0 (IBM SPSS, Inc., Chicago, IL, USA) was used for data analysis. Statistical significance was set at a *p*-value of <0.05.

## 3. Results

A sample of 40 participants was included in the study (BS, *n* = 21; MS, *n* = 19). Participants were mostly female (34 females: 85%) with a similar mean age, 21.0 (SD = 4.0) and 23.85 (SD = 1.61) for the BS and MS groups, respectively (Table 1). No significant differences were found between the BS and MS groups in gender and duration of CPR training, although most of the MS participants reported experience in training on advanced CPR advanced life support (χ^2^ = 4.912, *p* < 0.027).

Regarding the BLS questionnaire, there were significance differences in the number of correct answers between the groups (t = 2.334, *p* < 0.026). Likewise, CPR performance parameters were significantly higher in the MS group in comparison with the BS students (t = −2.307, *p* < 0.027), as shown in Appendix A. There were no statistically significant differences in the mean scores of stress and anxiety levels within and between the two groups, although a significant increment was observed in both scales’ values after simulation (Appendix A).

Table 2 shows the minimum, maximum and average temperature recordings as well as temperature increments obtained from the regions of interest in pre-test and post-test measurements. A characteristic thermographic profile represented by lower temperatures values in most of the facial regions was obtained following the simulation of all subjects (Figure 1). The analysis of pre-test–post-test temperature average values for the whole group showed statistically significant differences for all the selected facial regions: nose (t = 2.205, *p* < 0.033); forehead (t = 2.863, *p* < 0.007); periorbital (t = 2.420, *p* < 0.020); maxillary (t = 2.811, *p* < 0.008); and neck/upper chest (t = 2.953, *p* < 0.005).

The correlation analysis between the pre-test STAI scores and the facial temperature recordings showed positive and significant associations in the forehead area for both groups (maximum, BS, r = 0.627, *p* < 0.002; MS, r = 0.499, *p* < 0.03) and for the periorbital (maximum, r = 0.473, *p* < 0.042; average, r = 0.509, *p* < 0.026) and maxillary area (maximum, r = 0.537, *p* < 0.018; average, r = 0.534, *p* < 0.019) in the MS group (Appendix A). A statistically significant association was also observed between the post-test STAI scores and the temperature values in the BS group (neck and upper chest average, r = 0.559, *p* < 0.008). Regarding the entire group, positive and significant associations were observed for pre-test STAI scores with regard to both maximum and average temperature values for the following regions: forehead (maximum, r = 0.579, *p* < 0.000; average, r = 0.415, *p* < 0.004); periorbital (maximum, r = 0.394, *p* < 0.006; average, r = 0.318, *p* < 0.023); maxillary (maximum, r = 0.328, *p* < 0.019; average, r = 0.330, *p* < 0.019) and; neck area (maximum, r = 0.284, *p* < 0.038; average, r = 0.299, *p* < 0.030).

By correlating CPR performance parameters with the facial temperature values, a significant association was observed for the number of compressions in the following regions: periorbital area (temperature increment, r = 0.514, *p* < 0.017) in the BS group; nose (average, r = −0.524, *p* < 0.021) in the MS group and; maxillary region for both groups (minimum BS, r = 0.435, *p* < 0.049; minimum MS, r = 0.677, *p* < 0.001). At the same time, the correlation between the temperature gradient and the adequate compression rate showed a positive and significant association in the forehead area (minimum r = 0.445, *p* < 0.043) for the BS group, whilst the number of compressions with adequate depth and the mean compressions in 1 min were positively associated with the temperatures measured in the nose (r = 0.469, *p* < 0.043) and the neck (r = 0.537, *p* < 0.018) in the MS group (Appendix A). For the total of participants, the correlation of pre-test maximum and average temperatures with the number of compressions was statistically significant in the forehead (maximum, r = 0.372, *p* < 0.009; average, r = 0.447, *p* < 0.002), periorbital (maximum, r = 0.460, *p* < 0.001; average, r = 0.446, *p* < 0.002), and maxillary areas (maximum, r = 0.434, *p* < 0.003; average, r = 0.422, *p* < 0.003).

Multiple regression analysis showed a relationship between the pre-test maximum temperature recordings for all the facial regions and STAI pre-test scores (R^2^ = 0.395; F (5, 34) = 4.440; *p* < 0.003; d = 2.167), explaining 39.5% of the variance of STAI pre-test (Table 3). Significant regression equations were also obtained for the CPR global score (R^2^ = 0.378; F (5, 34) = 4.130; *p* <0.005; d = 1.890), number of compressions (R^2^ = 0.411; F (5, 34) = 4.751; *p* < 0.002; d = 2.087) and the compressions adequate rate (R^2^ = 0.282; F (5, 34) = 2.674; *p* < 0.038; d = 2.170).

## 4. Discussion

This work proposes a novel approach to evaluate the stress and anxiety perceived during simulation practices by means of facial infrared imaging analysis. The performance assessment of this methodology was proved throughout a cardiac arrest scenario by comparing differences in facial skin temperatures in bachelor and master nursing students.

First, it was intended to determine whether a temperature profile can be associated with the thermal facial variations occurring prior and following a simulation event. A specific temperature gradient was obtained depending on the facial region studied [21]. Our results showed a noticeable decrease in temperature recordings for the main facial regions, presenting statistically significant differences in average and maximum values. These findings were in agreement with previous studies, suggesting that changes in thermal facial gradients could be related to the thermoregulation of skin temperature experienced by individuals undergoing uncomfortable or stressful situations [21,22]. Hence, skin flow blood variations due to thermoregulatory vasodilatation and vasoconstriction are observed in response to thermal stress caused by either internal or external factors. Particularly, greater thermal variations have been reported in the nose temperature under stressful conditions in comparison with the cheekbone and the forehead [21]. A similar behavior was observed in our study when comparing temperature variations before and after simulation [29]. Likewise, higher thermal gradients were found in postgraduate students due to higher differences in stress and anxiety levels between pre-test and post-test measurements.

Therefore, it is worth considering the possibility of correlating temperature changes recorded in pre-test and post-test facial thermograms with the perceived stress and anxiety levels self-rated by simulation participants. Other studies have investigated facial temperature variations as the first body area that responds towards a stressful stimulus [24]. Nevertheless, to our knowledge, this is the first work that objectively quantify stress and anxiety levels during a simulation training by correlating infrared thermal facial imaging with validated scales. Our findings did not evidence statistically significant differences between thermal values and the stress VAS scores, probably due to the reduced stress response attributable to the low uncertainty of the simulation scenario [5]. However, statistically significant correlations were found in both groups between temperature gradients in specific facial regions and the anxiety scores evaluated by the STAI questionnaire. Specifically, maximum and average temperature values of the forehead region showed good correlation with STAI scores prior simulation for both groups. A positive correlation was also observed after the simulation practice in undergraduate students. These results confirmed that higher anxiety levels may be associated with the increase in the forehead temperature, as suggested by previous works [29,30,31,32]. Since the forehead area is one of the most stable temperature regions due to central vessel irrigation, thermal variations in this area are of great importance to provide information about stress diagnosis [30,33]. Likewise, maximum and average temperature values were positively correlated with the anxiety traits prior to the simulation practice [22]. These results were in line with previous studies since the increase in blood perfusion in the supraorbital vessels can also be related with stressful conditions [31]. Likewise, forehead temperature was expected to decrease after the simulation event, as reflected by the lower recordings obtained for both groups as a result of the psychophysiological response after cessation of the emotional stressor.

The tip of the nose is another critical area for the detection of human emotions due to its high sensitivity [29]. Different temperature patterns have been described when comparing experiences concerning high mental or cognitive load with anxiety or emotional stress. The former condition has been associated with increased nose temperature values, whereas a diminishing temperature profile has been reported for the latter [24,25,31,34]. A similar pattern of results involving lower temperature values in the tip of the nose with respect to other facial areas before the simulation event was found in our study. This is consistent with previous research since the emotional involvement of the task may be related to higher levels of stress and anxiety. However, different temperature gradients were observed when comparing the undergraduate and postgraduate groups following the simulation practice. The statistically significant drop in nose temperature values found in postgraduate students did not correspond with the slight increase observed in the other group. This fact may be related to the difference in perception of the physical and cognitive effort required for accomplishing the CPR performance among the groups, as the complexity of the task involves both emotional and mental load, which is probably more present in postgraduate students. As a result, the combination of both domains determines that the different temperature gradients did not have a statistically significant correlation with the anxiety perceived by participants.

In contrast, a consistent temperature profile was observed in the periorbital and maxillary areas by correlating pre-test STAI scores with maximum and average temperature values in the MS group. These facial regions, along with the forehead, area are considered good indicators of the activation of the sympathetic nervous system by triggering physiological responses under stressful situations. In this way, vasodilation, shivering and sweating result in reducing temperature values when the body responds to higher anxiety levels such as a simulation event. Accordingly, a good correlation between the temperature measurements of the above mentioned areas and the STAI scores was observed before the simulation practice [31,34].

Another key aspect that should be taken into account is the association between thermographic data and the effectiveness of CPR performance. Since the emotional and physio-psychological load may affect the simulation outcomes, monitoring of temperature facial variations can provide relevant information about the quality of CPR maneuvers. In this way, it is worth mentioning that CPR parameters related to the compression’s quality showed statistically significant correlations with temperature values in certain facial regions. For instance, the drop in the nose temperature values could be negative and significantly associated with worse CPR outcomes [35,36] as a result of higher anxiety levels, as previously described [24,25,31,34]. Similarly, higher quality compressions are associated with an increase in nose temperature values, as evidenced by the positive and significant correlation. With regard to the forehead area, the increase in temperature values can be positive and significantly correlated with a higher ratio of adequate chest compressions, thus indicating higher levels of mental load, cognitive effort or concentration [29], and thereby resulting in better CPR outcomes. In line with previous studies, the positive and significant correlation obtained when comparing the increase in maximum temperature values in the periorbital area with the number of compressions (BS group) indicates that temperature increments corresponding to higher concentration levels may lead to achieving adequate mean compressions without diminishing the compressions rate.

Although the interpretation of thermographic data did not yield statistically significant differences in all the facial regions depending on the participants’ experience, a consistent trend was found from the measurement of the temperature in the nose, forehead, maxillary periorbital and neck areas by examining the whole group of participants. The analysis of multiple regression models suggests that the impact of emotional factors can influence the temperature of the selected facial regions [24]. Skin temperature of the human face proved to be a significant predictor of perceived anxiety prior to a stressful stimulus, as shown by the significant regression equation found for the association between maximum temperature values and pre-test STAI scores [37]. Considering the quality of the simulation performance, facial temperature profiles also demonstrated to be good indicators of stress response as evidenced by the multiple regression analysis, thus proving the hypothesis enunciated above. Therefore, this pilot study offers an interesting perspective to recognize the effect of self-rated anxiety in controlled simulation-based experiences.

However, several limitations should be discussed. First, the comprehensive control of environmental conditions during thermographic measurements, such as the temperature and humidity variations, the distance between the participant and the examiner and the suitability of the participants clothing, need to be considered. The interpretation of thermographic information may also be improved by enhancing the selection of the facial regions of interest with supplementary software that is not fully dependent on the selection of pre-designed geometric shapes. Lastly, the recruitment and selection of participants was limited by convenience sampling. Likewise, the replicability of measurements with regard to intra- and inter-rater reliability and test–retest variations should be further considered.

## 5. Conclusions

The presence of stress and anxiety during simulation-based learning may affect the performance outcomes. This study takes advantage of infrared thermal imaging to study the relationship between differences in facial skin temperature and the perception of anxiety throughout a cardiac arrest simulated scenario. The analysis of facial temperature variations showed good correlations with either the anxiety scale or standard quality resuscitation parameters, showing consistent thermographic profiles for the forehead, maxillary and periorbital areas. Consequently, the utilization of facial temperature values should be taken into consideration to predict the influence of anxiety during simulation training. Despite being a pilot study, the results are expected to improve assessment performance prior to a simulation practice by providing valuable information on the anxiety traits of simulation participants. Further research is needed to examine the reliability of infrared imaging technology as a valid screening tool for the objective quantification and diagnosis of emotional and cognitive load in simulation training practices.

## Figures and Tables

**Figure 1 healthcare-08-00206-f001:**
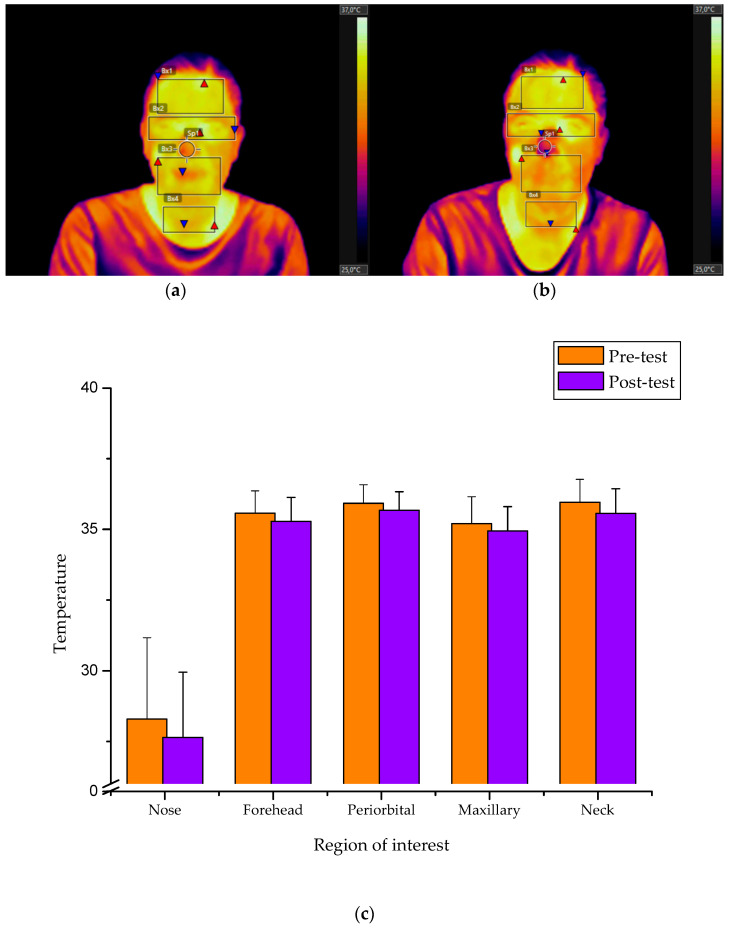
Infrared thermograms showing maximum (red triangles) and minimum temperature (blue triangles) gradients of the selected regions of interest: (**a**) prior and (**b**) following the simulation event. (**c**) Representation of pre-test–post-test average temperatures depending on the facial area for the total of participants.

**Table 1 healthcare-08-00206-t001:** Comparison of demographics for the undergraduate group versus the postgraduate group.

Sociodemographic Characteristics		Undergraduate Bachelor Students (BS)		Postgraduate Master Students (MS)	Statistic Values χ^2^/t	*p* Value
		*N* = 21 mean ± SD (%)		*N* = 19 mean ± SD (%)		
**Sex**	Female	18 (85.7)		16 (84.2)	0.018 ^b^	0.894 ^b^
	Male	3 (14.3)		3 (15.8)		
**Age**		21.0 (4)		23.85 (1.61)	−2.890 ^a^	0.006 *^,a^
**Educational level**	Baccalaureate	17 (81)		0		0.000 *^,b^
	Professional training	3 (14.3)		0	36.190 ^b^	
	Other Bachelor of Science	1 (4.8)		19 (100)		
**Practicum in special health services**	Yes	0		19 (100)		0.000 *^,b^
	No	21 (100)		0	40.0 ^b^	
**Number of special health services in practicum**		0		1.84 (0.83)	−9.625 ^a^	0.000 ^a^
**Work in special health services**	Yes	0		10 (52.6)	14.737 ^b^	0.000 *^,b^
	No	21 (100)		9 (47.4)		
**Number of special health services working**	0	0		0.84 (1.05)	−3.618 ^a^	0.002 *^,a^
**Training on basic CPR (basic life support)**	Yes	Last two years	1 (4.8)	6 (31.6)	6.686 ^b^	0.010 *^,b^
		More than two years	2 (9.5)	2 (10.5)		
	No		18 (85.7)	9 (47.4)		
**Duration basic CPR training**		37.67 (46.11)		56.33 (37.67)	6.750 ^b^	0.455 ^a^
**Training on advanced CPR (advanced life support)**	Yes	0		4 (21.1)	4.912 ^b^	0.027 *^,b^
	No	21 (100)		15 (78.9)		

* *p* < 0.05; ^a^ t-Student independent samples; ^b^ chi-squared Pearson; SD: Standard Deviation.

**Table 2 healthcare-08-00206-t002:** Temperature values of the regions of interest for the undergraduate group versus the postgraduate group.

Facial Region	Temperature Value	Moment	Temperature Mean (SD)	Undergraduate Bachelor Students (BS)		Temperature Mean (SD)	Postgraduate Master Students (MS)		t	*p* Value Groups
				t-Paired	Significance		t-Paired	Significance		
**Nose**	Average	Pre-test	27.87 (2.56)	−1.014	0.323	28.76 (3.20)	4.095	0.001 **	−0.972	0.337
		Post-test	28.17 (2.31)			27.06 (2.21)			−0.961	0.129
		Difference	0.30 (1.36)		-	−1.70 (1.81)		-	1.553	0.000 *
**Forehead**	Maximum	Pre-test	35.79 (0.76)	0.982	0.338	35.79 (0.76)	2.362	0.030 *	1.557	0.072
		Post-test	35.59 (0.89)		-	34.95 (0.66)			3.980	0.014 *
		Difference	−0.20 (0.91)		-	−0.39 (0.72)		-	3.923	0.462
	Average	Pre-test	34.90 (0.77)	1.291	0.211	34.13 (1.34)	2.939	0.009 **	1.851	0.031 *
		Post-test	34.60 (1.10)			33.52 (1.26)			1.849	0.006 *
		Difference	−0.30 (−0.30)			−0.61 (0.91)			2.578	0.317
	Minimum	Pre-test	32.62 (1.64)	−0.432	0.671	30.40 (2.75)	0.789	0.440	2.617	0.005 *
		Post-test	32.77 (1.03)			30.09 (2.54)			0.743	0.000 *
		Difference	0.15 (1.57)			−0.31 (1.72)			0.752	0.383
**Periorbital**	Maximum	Pre-test	35.91 (0.63)	1.142	0.267	35.93 (0.70)	2.560	0.020 *	2.247	0.935
		Post-test	35.67 (0.64)			35.57 (0.67)			2.190	0.346
		Difference	−0.15 (0.59)			−0.36 (0.62)			2.906	0.267
	Average	Pre-test	34.01 (0.90)	0.294	0.771	33.93 (0.82)	3.670	0.002 **	2.886	0.764
		Post-test	33.96 (0.85)			33.35 (0.59)			1.013	0.012 *
		Difference	−0.05 (0.81)			−0.58 (.69)			1.021	0.033 *
	Minimum	Pre-test	28.84 (2.10)	0.366	0.718	29.28 (2.07)	3.596	0.002 **	3.137	0.509
		Post-test	28.73 (1.77)			28.18 (1.66)			3.062	0.323
		Difference	−0.11 (1.43)			−1.10 (1.33)			4.443	0.030 *
**Maxillary**	Maximum	Pre-test	35.29 (0.99)	0.579	0.569	35.11 (0.94)	2.109	0.049 *	4.282	0.548
		Post-test	35.177 (0.90)			34.70 (0.73)			0.883	0.074
		Difference	−0.11 (0.90)			−0.41 (0.85)			0.879	0.294
	Average	Pre-test	33.27 (1.30)	1.285	0.214	33.10 (1.25)	2.872	0.010 *	−0.082	0.681
		Post-test	32.93 (1.08)			32.42 (1.12)			−0.082	0.153
		Difference	−0.34 (1.21)			−0.68 (1.03)			0.953	0.345
	Minimum	Pre-test	27.59 (2.25)	0.287	0.777	28.20 (2.55)	4.011	0.001 **	0.951	0.424
		Post-test	27.46 (1.74)			26.57 (2.08)			1.126	0.148
		Difference	−0.12 (1.98)			−1.63 (1.77)			1.123	0.016 *
**Neck/** **Upper chest**	Maximum	Pre-test	36.05 (0.92)	2.177	0.042 *	35.85 (0.71)	2.189	0.042 *	0.302	0.436
		Post-test	35.652 (0.99)			35.47 (0.74)			0.304	0.514
		Difference	−0.40 (0.84)			−0.38 (0.75)			2.635	0.934
	Average	Pre-test	34.50 (0.91)	1.711	0.103	34.23 (0.62)	2.547	0.020 *	2.681	0.279
		Post-test	34.21 (0.93)			33.84 (0.62)			2.210	0.158
		Difference	−0.30 (0.79)			−0.38 (0.66)			2.228	0.703
	Minimum	Pre-test	31.21 (1.91)	−0.054	0.957	30.69 (2.01)	0.763	0.455	−0.667	0.407
		Post-test	31.23 (1.65)			30.17 (1.93)			−0.668	0.069
		Difference	0.02 (2.01)			−0.51 (2.94)			1.001	0.499

* *p* < 0.05, ** *p* < 0.01; t-Student paired samples; t independent samples.

**Table 3 healthcare-08-00206-t003:** Multiple linear regression analysis to model the relationship between the State-Trait Anxiety Inventory (STAI) pre-test scores and maximum temperature recordings in the selected facial regions before simulation.

Dependent Variable: STAI Pre-Test	Unstandardized Coefficients		Standardized Coefficients
	B	Standard Error	Beta
Constant	−63.284	37.913	
Nose temperature	0.182	0.278	0.107
Forehead temperature	6.555	1.826	1.056
Periorbital temperature	−2.310	2.096	−0.309
Maxillary temperature	−0.806	1.175	−0.157
Neck/upper chest temperature	−1.117	1.098	−0.187

B is the unstandardized coefficient beta.

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
