# Peer review of "Effect of Facial Skin Temperature on the Perception of Anxiety: A Pilot Study"

_healthcare, 2020, doi:10.3390/healthcare8030206_

Round 1
Reviewer 1 Report
The strengths is a very precise measurement of temperature in the facial skin and a very good statistical evaluation. The problem of presentation is that at a first glance the reader looks at the picture and diagram and then he sees a diagram with minimal differences because of the use of an absolute scale of temperature and then he reads of very high significances between groups. So, to lead the reader it is better to offer diagrams with indicators of significances between groups e. g. undergraduates vs. postgraduates and so on. That means make flashy diagrams for all parameters because the tables of the significances are too overloaded and can be presented in an addendum. Then these good results are presented adequate.
Please present another temperature diagram - replace or magnify the temperature differences and significances in the existing - otherwise the differences look not very convincing at a first glance.
Minor error: line 33 in Introduction...such as...(not us).
Reviewer 2 Report
I would have liked more clarity regarding the study design. As a new reader, I asked myself ‘is this a correlational study?”, “is this an intervention study?”. The aims of the study (lines 66-67) implied that it was a correlational study looking at possible associations, whereas later it was described as quasi-experimental, with dependent and independent variables being outlined. At various points, the authors imply causality (e.g. line 222), - and I would have liked more of a discussion of this claim, including the limitations and alternative explanations being suggested for any apparent causal association.
Author Response
We would like to thank the reviewers for their constructive and valuable suggestions. Their careful reading has contributed to improving the quality of this review. According to their comments, the manuscript has been comprehensively revised and all their requests have been addressed. Following the editor instructions, we have included a point by point response to the reviewers’ comments and the changes made have been highlighted accordingly in the revised version of the manuscript.
In order to help reviewing amendments done to the manuscript, below we reproduce in full the reviewer’s comments (black color and italics font) and immediately after each comment, the reply provided in red color and portions of the revised text in the paper between quotation marks (red color and italics).
Reviewer 2
I would have liked more clarity regarding the study design. As a new reader, I asked myself ‘is this a correlational study?”, “is this an intervention study?”. The aims of the study (lines 66-67) implied that it was a correlational study looking at possible associations, whereas later it was described as quasi-experimental, with dependent and independent variables being outlined. At various points, the authors imply causality (e.g. line 222), - and I would have liked more of a discussion of this claim, including the limitations and alternative explanations being suggested for any apparent causal association.
We thank the reviewer for the careful reading and positive comments.
Regarding the study design, as the reviewer accurately noticed, we used ‘a pretest-posttest quasi-experimental design’ (line 73). Likewise, the aim of the study was ‘to examine the relationship between differences in skin temperature of different facial regions and the stress and anxiety perceived by Bachelor and Master nursing students during a cardiac arrest simulated based-scenario’ (lines 65-67). Therefore, we focus on the measurement of temperature variations as dependent variables after participation in a clinical simulation experience. Previous simulation based-learning interventions have employed quasi-experimental designs by making use of the comparison of several groups, without taking advantage of a correlational research approach (Mills et al., Nurse Education Today 45 (2016) 9–15; Fernández-Ayuso et al. JMIR Serious Games 2018 [17]; Plemmons et al., Nurse Education Today 62 (2018) 107–111[6]; Kim, Nurse Education Today 61 (2018) 258–263). In this work, we use a convenience sample wherein participants were divided in two groups depending on their educational level. However, the main objective of the study was to obtain a temperature profile after the intervention. In this sense, our results represent either the results for all participants or the differences between groups.
Nevertheless, following the reviewer’s recommendation, to clarify this point, we have replaced ‘between’ by ‘in’ (line 222): ‘The performance assessment of this methodology was proved throughout a cardiac arrest scenario by comparing differences in facial skin temperatures in Bachelor and Master nursing students’.
Additionally, the statement on the explanation of possible associations is mentioned in line 287: ‘Although the interpretation of thermographic data did not yield statistical significance differences in all the facial regions depending on the participants’ experience, a consistent trend was found from the measurement of the temperature in the nose, forehead, maxillary periorbital and neck areas by examining the whole group of participants.’
Finally, the selection of the sample has been considered in the limitations of the study:
‘Lastly, the recruitment and selection of participants was limited by convenience sampling. Likewise, the replicability of measurements with regard to intra- and interrater reliability and test-retest variations should be further considered’.
Reviewer 3 Report
In the present pilot study, The Authors aimed to evaluate the relationships between differences in skin temperature of different facial regions and the stress and anxiety perceived by Bachelor and Master nursing students during a cardiac arrest simulated based-scenario. The Authors' hypotheses were that: (i) thermal facial gradients obtained by infrared thermal imaging can be associated with the scores assessed by stress and anxiety validated questionnaires and (ii) higher anxiety levels will result in poorer clinical performance with regard to temperature variations in certain facial regions.
Overall, I found the present study timely, very interesting, well conducted and scientifically sound. I have some suggestions aimed to improve the quality of the paper and these are outlined below:
1) In the Introduction, it should be specified that the perception of the anxiety may be also influenced by several personality traits and neuropsychological conditions often related to pharmacological treatments. This point should be briefly discussed with appropriate references such as Tempesta et al. Prog Neuropsychopharmacol Biol Psychiatry. 2013; Caldirola et al. Compr Psychiatry. 2014; De Berardis et al. Arch Suicide Res. 2017).
2) The recruitment process was developed during the lecture time and their participation was voluntary and did not include remuneration or course credit. But, on which basis the sample was selected? Please, add some more informations.
3) In the Tables, I suggest to add also the statistical values and effect sizes rather than simply the p.
4) I believe that table 3 (bivariate correlations) is unnecessary as it can be eliminated and condensed as text in results.
5) Concerning the multiple regression analyses, please add the Durbin Watson coefficient.
6) Concerning limitations, please add the this was a convenience sample.
